# Rare Recombinant GI.5[P4] Norovirus That Caused a Large Foodborne Outbreak of Gastroenteritis in a Hotel in Spain in 2021

María Ester Alarcón-Linares,[a] Antonio Moreno-Docón,[b,c] Lorena Pérez-Serna,[a] Juan Camacho,[d] Diego Sánchez Rodriguez,[e] María Luisa Gutiérrez-Martín,[a] Antonio Broncano-Lavado,[f] Juan-Emilio Echevarria,[d,g] María Cabrerizo,[d,g] María D. Fernández-García[d,g]

[a]Murcia Regional Health Council, Murcia, Spain
[b]Hospital Universitario Virgen de la Arrixaca, Murcia, Spain
[c]IMIB—Arrixaca, Universidad de Murcia, Murcia, Spain
[d]Centro Nacional de Microbiología, Instituto de Salud Carlos III, Madrid, Spain
[e]Unidad Docente de Medicina Preventiva y Salud Pública, Murcia, Spain
[f]Instituto de Investigación Sanitaria—Fundación Jiménez Díaz Hospital Universitario, Universidad Autónoma de Madrid (IIS-FJD, UAM), Madrid, Spain
[g]CIBER de Epidemiología y Salud Pública (CIBERESP), Instituto de Salud Carlos III (ISCIII), Madrid, Spain

**ABSTRACT**    Noroviruses are among the most important causes of acute gastroenteritis (AGE). In summer 2021, a large outbreak of norovirus infections affecting 163 patients, including 15 norovirus-confirmed food handlers, occurred in a hotel in Murcia in southeast Spain. A rare GI.5[P4] norovirus strain was identified as the cause of the outbreak. The epidemiological investigation determined that norovirus transmission might have been initiated through an infected food handler. The food safety inspection found that some symptomatic food handlers continued working during illness. Molecular investigation with whole-genome and ORF1 sequencing provided enhanced genetic discrimination over ORF2 sequencing alone and enabled differentiation of the GI.5[P4] strains into separate subclusters, suggesting different chains of transmission. These recombinant viruses have been identified circulating globally over the last 5 years, warranting further global surveillance.

**IMPORTANCE**    Due to the large genetic diversity of noroviruses, it is important to enhance the discriminatory power of typing techniques to differentiate strains when investigating outbreaks and elucidating transmission chains. This study highlights the importance of (i) using whole-genome sequencing to ensure genetic differentiation of GI noroviruses to track chains of transmission during outbreak investigations and (ii) the adherence of symptomatic food handlers to work exclusion rules and strict hand hygiene practices. To our knowledge, this study provides the first full-length genome sequences of GI.5[P4] strains apart from the prototype strain.

**KEYWORDS**    norovirus, whole-genome sequencing, recombination, outbreak, foodborne, food handler, genotyping, genetic recombination

Address correspondence to María D. Fernández-García, mdfernandez@isciii.es.

The authors declare no conflict of interest.

Noroviruses are among the most important causes of acute gastroenteritis (AGE) globally. The virus has a fecal-oral transmission route and may be spread by direct person-to-person contact or indirectly via contaminated food or water. Strains mostly associated with human infection belong to genogroups I, II, and IV, with GII being the most prevalent (1). The genetic diversity is further increased through frequent intergenotype recombination events, with breakpoints most often identified at the junction of the ORF1 (encoding RNA-dependent RNA polymerase) and the ORF2 (encoding the major capsid protein VP1) of the norovirus genome (2).

**Epidemiological investigation.** Between 21 July and 22 August 2021, an outbreak of AGE was reported in a hotel in Murcia in southeast Spain. The Regional Public Health Department, in collaboration with the regional Food Safety Authority, carried out an investigation to identify the source and implement control measures. In total, 163 symptomatic cases were reported: 156 clients and 7 food handlers. Table S1 in the supplemental material shows the clinical and demographic characteristics of cases associated with the outbreak. Symptoms were self-limited, and there were no hospitalizations. All but one case had full board, and thus they were living and having meals inside the hotel. The first known infections occurred on 21 July and involved a probable symptomatic client and a symptomatic laboratory-confirmed food handler (Fig. 1). The next day, five food handlers reported gastrointestinal symptoms, supporting the hypothesis that they could have contributed to the initial spread of the virus. Based on the shape of the epidemiological curve, we hypothesized that the outbreak might have been further propagated by both direct person-to-person contact and direct contact with contaminated surfaces. The food safety inspection conducted on 26 July found that some symptomatic food handlers continued working during their illness. On 29 July, all food handlers were restricted temporarily from their duties. On 4 August, the hotel closed the kitchen and all food and drink services. On 18 August, additional cases were reported. A retrospective cohort study calculating the attack rates for individual food items served at the hotel's restaurant was conducted to identify the source of infection. The cohort study recruited 65 probable primary cases and 14 controls who completed the questionnaire, but in the statistical analysis, no specific food items were associated with illness (Table S2).

**Microbiological investigation.** A total of 17 stool samples (from 6 symptomatic and 8 asymptomatic food handlers and 3 symptomatic clients) were positive for norovirus GI. For symptomatic patients, samples were collected between 0 and 5 days after the onset of symptoms. Table S3 describes the samples collected for laboratory testing during the outbreak investigation. Sixteen norovirus GI-positive stool samples were submitted to the National Centre of Microbiology for genotyping. The samples were tested for norovirus GI using the dual polymerase-capsid genotyping strategy (3) and Sanger sequencing. For genotyping, we used the Norovirus Typing Tool (https://www.rivm.nl/mpf/typingtool/norovirus/). Nucleotide sequencing was successful in 15/16 samples (93.7%). All sequenced strains belonged to genotype GI.5[P4]. The sequences were aligned using ClustalW software. Phylogenetic trees were inferred from study sequences (only those with >450 nucleotides [nt]; $n$ = 14) and representative GI.5[P4], GI.5[P5], and GI.4[P4] ORF1 and ORF2 sequences available at GenBank.

A phylogenetic tree of the partial polymerase sequences (ORF1) showed that the GI.5[P4] outbreak strains found in this study were segregated into two clearly distinct genetic subclusters, I and II (Fig. 2A), that cocirculated during the outbreak, suggesting different chains of transmission. The GI.5[P4] outbreak strains in subcluster I clustered with worldwide GI.5[P4] strains; most were collected in Asia between 2018 and 2021, and two were collected in Spain between 2018 and 2019. The GI.5[P4] outbreak strains in subcluster II formed a monophyletic branch with a bootstrap value of 93% and an intracluster nucleotide identity of >99.93%. This high degree of sequence similarity suggests that the infection probably had a common source or was spread via person-to-person transmission. Segregation into different subclusters was not observed in the phylogenetic tree for the 5' end of ORF2 (Fig. 2B).

To further study the phylogenetic relationships, complete genomic sequences were obtained from randomly selected outbreak strains, two from each subcluster, as previously described (4). There was 100% agreement between the genotypes assigned by Sanger sequencing and whole-genome sequencing (WGS). Phylogenetic analysis of the complete ORF1 and ORF2 regions confirmed that the outbreak GI.5[P4] sequences represented distinct clades with bootstrap values of 99% to 100% (Fig. 2C and D).

To characterize the potential recombination events of the GI.5[P4] strains identified in this study, a similarity plot analysis was performed by aligning the complete genome sequences of the study strains with those of closely related types, GI.5[P5] and GI.4[P4].

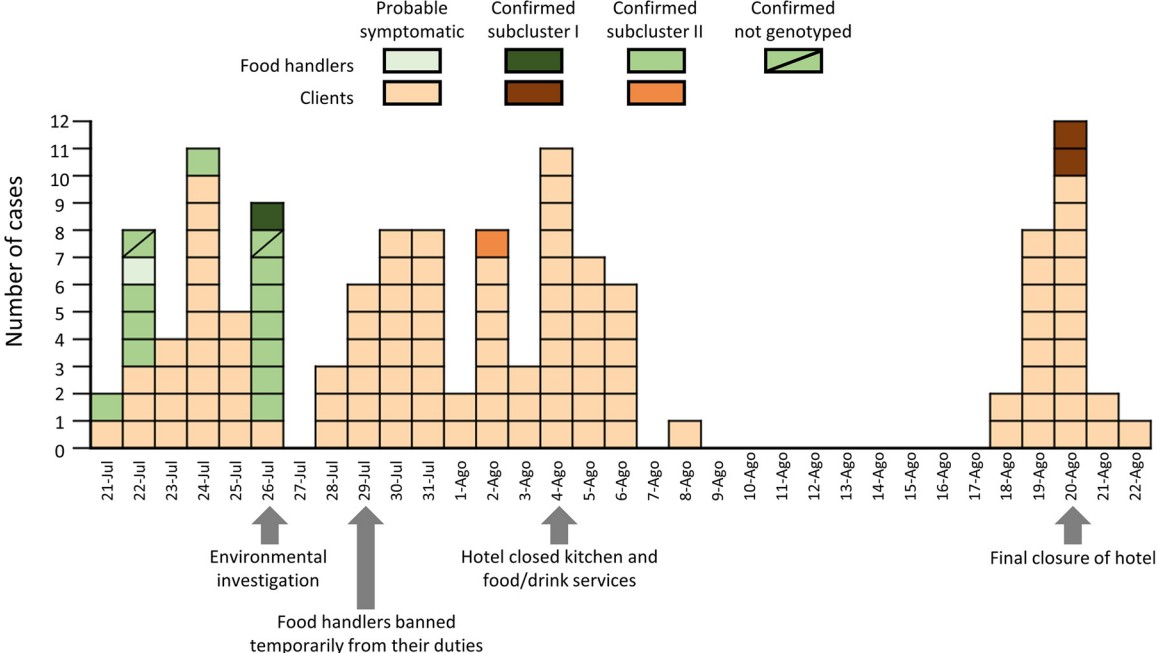

**FIG 1** Epidemiological curve of cases associated with the GI.5[P4] norovirus foodborne outbreak by date of illness onset. Probable cases with a known illness onset date were included (*n* = 110). For confirmed cases (*n* = 17), we used the illness onset date, except for the asymptomatic food handlers (*n* = 8), for which the sample collection date was used. Arrows indicate important events in the outbreak investigation. Subsequent genetic analysis of the GI.5P[4] strains identified them as members of subclusters I and II.

While the ORF1 presented high identity with that of the GI.4[P4] norovirus identified in the Netherlands in 2011, the ORF2 and ORF3 presented high identity with those of GI.5 [P5] norovirus identified in Argentina in 2012 (Fig. S1). Therefore, the GI.5[P4] was identified as an intergenotype recombinant of the GI.4[P4] and GI.5[P5] strains.

GI.5[P4] is a rarely reported norovirus genotype, with strains sporadically reported worldwide in the last years. A 5-year surveillance study across 6 continents in AGE cases describes the detection of only 10 GI.5[P4] strains, collected in 2018 and 2019 in Nicaragua (*n* = 9) and Hong Kong (*n* = 1) (5). Moreover, 23 partial GI.5[P4] sequences are available at GenBank. According to the GenBank data, GI.5[P4] was first detected in India in 2016 and was subsequently detected in Brazil, China, South Korea, and Spain between 2018 and 2021. Only one previous outbreak of GI.5[P4] in a nursing home in 2019 affecting 18 individuals has been reported in Spain (6). In our study, we report a close genetic relationship between all reported GI.5[P4] strains (>99% nucleotide and amino acid similarity), which suggests that this genotype is an emergent virus that has disseminated globally through Asia, the Americas, and Europe within the last 5 years.

Due to the broad genetic diversity of noroviruses, with multiple types and variants cocirculating at the same time, it is important to enhance the discriminatory power of typing techniques to differentiate strains when investigating outbreaks and elucidating transmission chains. In this study, we show that WGS or partial ORF1 sequencing provided enhanced genetic discrimination of similar GI strains over 5'-end ORF2-based genotyping alone. This is important since many of the laboratories from the NoroNet network (Europe, Asia, Oceania, and Africa) use methods targeting only the 5' side of ORF2, a region that has been recommended for routine genotyping in epidemiological studies because it is highly conserved (7–9). Our results support the recommendation of using WGS, or at a minimum, a protocol including the ORF1 and the ORF2 overlap, when tracking outbreaks and chains of transmission in order to detect recombinant strains while ensuring good genetic differentiation between the strains (7, 10, 11).

Since only a low infectious dose of norovirus is needed to cause infection (12), outbreaks can easily occur. Although the specific source of this outbreak remains unclear, based on the epidemiological investigation, norovirus transmission might have been

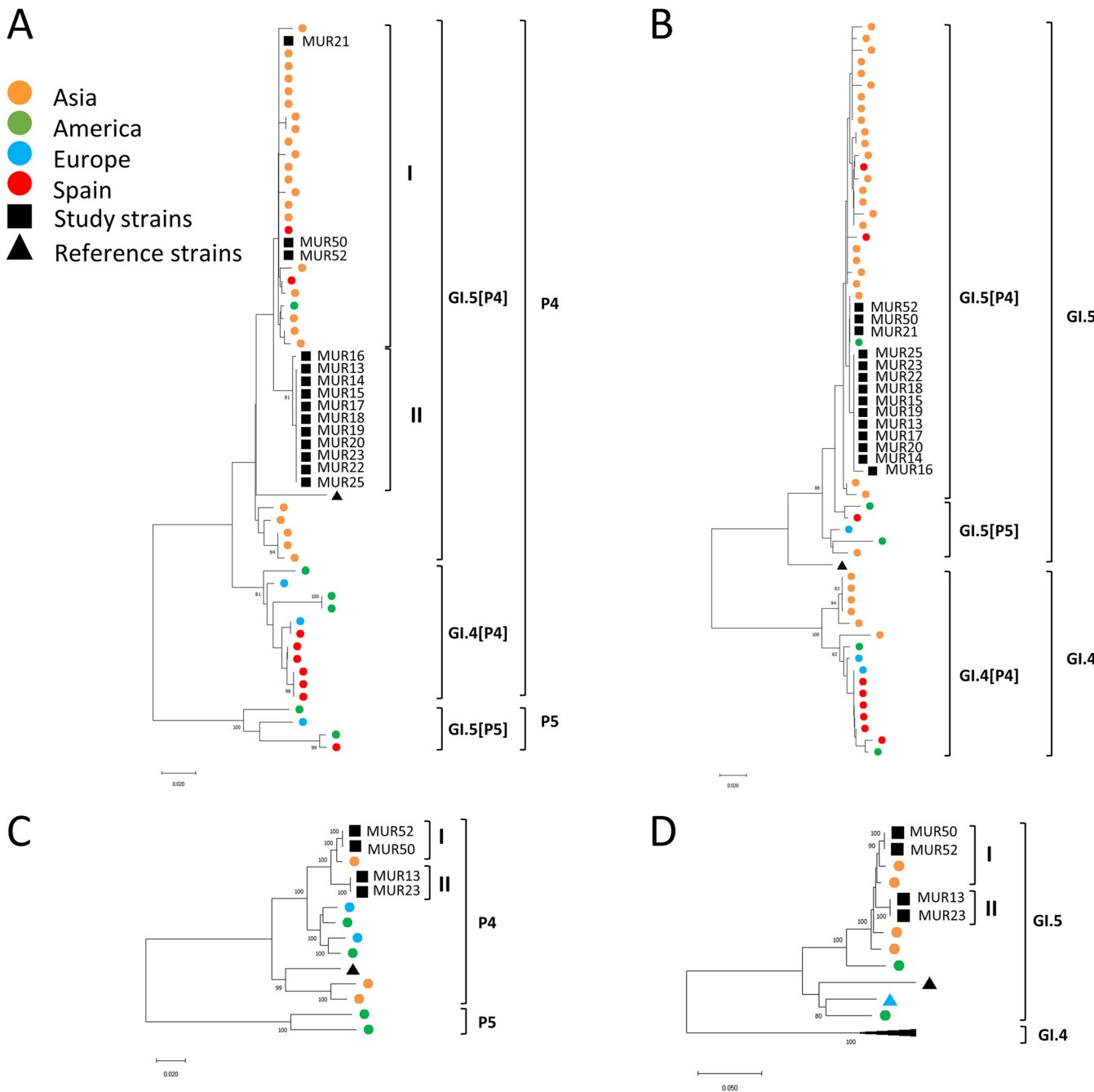

**FIG 2** Phylogenetic analysis of the norovirus genomes identified in this study based on a 250-bp partial 3'-end sequence of ORF1 (A), a 274-bp partial 5'-end sequence of ORF2 (B), a complete ORF1 sequence (C), and complete ORF2 sequence (D). The sequences of closely related types (GI.4[P4] and GI.5[P5]) included in the analysis were those with highest scores when using the genomic regions ORF1 and ORF2 as queries for BLASTn (https://blast.ncbi.nlm.nih.gov/Blast.cgi). Nucleotide sequences were analyzed using MEGA 5.0 software (http://megasoftware.net/) with the neighbor-joining method. Bootstrap values greater than 80% are shown. The robustness of the nodes was tested using 1,000 bootstrap replications. Scale bars indicate the number of nucleotide substitutions per site. Noncollapsed trees with taxon names can be found in Fig. S2A to D in the supplemental material.

initiated through food that had been manually handled by an infected food handler. It is worth noting that a similar norovirus outbreak took place in the same hotel 2 years before, in 2019 (13). Symptomatic food handlers were also involved, although the causative pathogen then was norovirus GII. This emphasizes the importance of the immediate exclusion of symptomatic food handlers, adherence of symptomatic food handlers to work exclusion rules, strict hand hygiene practices, and decontamination of environmental surfaces to prevent contamination of food items.

In summary, we show that it might be difficult to distinguish GI norovirus strains that

belong to different chains of transmission in an outbreak based only on the short genomic region of the 5′ end of ORF2, while WGS or partial ORF1 sequencing both provide a higher resolution. Because the rare recombinant genotype GI.5[P4] has caused a large outbreak and has spread quickly on multiple continents over the last years, it warrants further global surveillance. A sustained and real-time public health laboratory capacity for norovirus genotyping using WGS during outbreaks, paired with epidemiological data, would offer the opportunity to trace chains of transmission, confirm cases, and distinguish outbreak-related cases from sporadic cases.

**Ethical statement.** Ethics approval was not required, as the clinical isolates were collected as part of routine surveillance and outbreak system activities.

**Data availability.** Sequences were submitted to GenBank under the accession numbers OP709668 to OP709677. Whole-genome sequences were submitted under the accession numbers OP723153 to OP723156.

## SUPPLEMENTAL MATERIAL

Supplemental material is available online only.
**SUPPLEMENTAL FILE 1**, PDF file, 0.3 MB.

## ACKNOWLEDGMENT

We thank the Genomics and Bioinformatic Departments at the ISCIII for technical assistance. This study was partially funded through project PI20CIII/00005.

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
