## [Reviewer comments · Microbiology Spectrum]

Microbiology Spectrum

Rare Recombinant GI.5[P4] Norovirus causing a large foodborne outbreak of gastroenteritis in a hotel, Spain, 2021

MARIA ESTER ALARCON-LINARES, Antonio Moreno-Docón, Lorena Perez-Serna, Juan Camacho, Diego Sanchez-Rodriguez, Maria Luisa Gutierrez-Martin, Antonio Broncano-Lavado, Juan Echevarría, M Cabrerizo, and Maria Dolores Fernandez-Garcia

Corresponding Author(s): Maria Dolores Fernandez-Garcia, Instituto de Salud Carlos III

Review Timeline:

Submission Date:	November 26, 2022
Editorial Decision:	January 3, 2023
Revision Received:	January 17, 2023
Editorial Decision:	January 18, 2023
Revision Received:	January 20, 2023
Accepted:	January 25, 2023

Editor: Gabriel Parra

Reviewer(s): The reviewers have opted to remain anonymous.

Transaction Report:

DOI: <https://doi.org/10.1128/spectrum.04857-22>

January 3, 2023

Dr. Maria Dolores Fernandez-Garcia
Instituto de Salud Carlos III
Centro Nacional de Microbiología
Majadahonda, Madrid
Spain

Re: Spectrum04857-22 (Rare Recombinant GI.5[P4] Norovirus causing a large foodborne outbreak of gastroenteritis in a hotel, Spain, 2021)

Dear Dr. Maria Dolores Fernandez-Garcia:

Happy New Year and thank you for submitting your manuscript to Microbiology Spectrum. Your paper was reviewed by two experts in norovirus epidemiology and both agreed that this report is of interest. Please see their comments below.

Link Not Available

Sincerely,

Gabriel Parra

Journals Department
Reviewer comments:

Reviewer #1 (Comments for the Author):

In this manuscript, authors investigated food-borne outbreaks that occurred in a hotel in Murcia, Spain. During the microbiological investigation using specimens from symptomatic and asymptomatic food-handlers and symptomatic clients, authors identified rare human norovirus GI.4[P5] as a source of the outbreak. Genetic analyses concluded that this norovirus spread by at least two transmission chains in the hotel.

Overall, this manuscript is well written and I agree with the authors that molecular analyses using whole-genome sequence data, not a short fragment on the conserved regions, could elucidate high resolution of transmission dynamics during the

norovirus outbreaks. Although epidemiological analysis failed to identify specific food items with high risk of symptomatic infections, as norovirus was detected from an index case, a symptomatic food handler, I agree that this is a food-borne outbreak caused by norovirus contaminated from food. Please find below minor suggestions to improve the manuscript.

1. I agree that this could be a food-borne outbreak, but have you conduct risk analyses with those exposed and unexposed to any food/drink in the hotel? Authors analyzed for individual food/drink items but is there any overall differences in relative risk between those exposed and unexposed to any items? This analysis may enable to confirm if the virus spread by person-person or by food/use of restaurant.
2. There are some food items that showed statistically significant high risk with $p < 0.05$. I would recommend to briefly explain/discuss about these items in the text.
3. Please explain what kind of specimens and when they were collected (e.g. days after the onset) during the microbiological investigation.
4. As norovirus typing tool provides both capsid and polymerase type information and discrepancy between capsid and polymerase type already indicates that this strain (GI.5[P4]) is a recombinant virus with GI.5 and GI.P4 virus, I would suggest to omit the simplot and recombination analyses from the manuscript.
5. Please provide legend for the Supplementary Figure 1.
6. Table S1: Please indicate if the blank (symptom in second part of the outbreak) indicates 0 case or no data available.
7. Lines 116-120: I found several typos for GI.5[P4]. e.g., GI5[P4], GI.5 [P4]. No period or extra space before the bracket.

Reviewer #2 (Comments for the Author):

In this study, researchers investigated instances of norovirus infection among the guests and food-handlers at a hotel in the Summer of 2021. The majority of the samples sequenced was of GI.5P[4]. The authors performed a thorough survey of foods eaten by guests during the outbreak period, but no definitive foodsource was implicated. Transmission by fomite or food-handlers was also a possibility.

Major comments:

1. Out of 16 sequenced samples, 15 were GI.5P[4]. It would be good to mention in the manuscript what genotype the remaining one was.
2. Line 64, "The first confirmed case, with symptom onset on July 21, was one food-handler". Looking at Figure 1, this sentence may appear confusing because Figure 1 showed one probable symptomatic client (light brown) and one confirmed subcluster II (light green). The authors referred to "subcluster II" as a genetic grouping, which not introduced until Figure 2. I recommend modifying this sentence to something like "The first known infection occurred on July 21 involving a probable symptomatic client and a symptomatic laboratory-confirmed food-handler." To reduce confusion, the authors may want to add something like "Subsequent genetic analysis of the GI.5P[4] identified the strains as members of subcluster I and II" in the Figure 1 legend.
3. Figure 1. I counted 127 squares, but the legend mentioned $n=110$ known onset date and $n=8$ collection date. Why the discrepancies? Additionally, there were 11 confirmed food-handlers and 1 client infected with subcluster II, yielding 12 genotyped samples, but only 11 strains shown on the phylogenetic tree?
4. Line 79. The authors wrote "A total of 18 samples (from 7 symptomatic and 8 asymptomatic food-handlers, 3 symptomatic clients) were positive for norovirus GI. Supplementary table 3 describes samples for laboratory testing during the outbreak investigation." However, this table indicates that there were only 6 symptomatic food-handlers positive for GI. Why?
5. Supplementary Figure 1 is so hazy it's not even legible from the source file. Please ensure the quality of the figure.
6. Chhabra et al., 2019 established the prototype strains for the purpose of genotyping. It is strongly encouraged that the phylogenetic trees include the necessary prototypic strain (AB042808 Chiba407 for RdRp, and AJ277614 Musgrove for VP1).
7. At least 8 references were missing volumes and page numbers.

Staff Comments:

Preparing Revision Guidelines

Please return the manuscript within 60 days; if you cannot complete the modification within this time period, please contact me. If you do not wish to modify the manuscript and prefer to submit it to another journal, please notify me of your decision immediately so that the manuscript may be formally withdrawn from consideration by Microbiology Spectrum.

Ref.: Spectrum04857-22

Rare Recombinant GI.5[P4] Norovirus causing a large foodborne outbreak of gastroenteritis in a hotel, Spain, 2021

The authors would like to thank the reviewers for their comments. We believe that their input has strengthened the presentation and content of the manuscript. We have addressed their comments as follows (all the amends on your manuscript are indicated by highlighting in yellow the changes):

Reviewer #1:

In this manuscript, authors investigated food-born outbreaks that occurred in a hotel in Murcia, Spain. During the microbiological investigation using specimens from symptomatic and asymptomatic food-handlers and symptomatic clients, authors identified rare human norovirus GI.4[P5] as a source of the outbreak. Genetic analyses concluded that this norovirus spread by at least two transmission chains in the hotel.

Overall, this manuscript is well written and I agree with the authors that molecular analyses using whole-genome sequence data, not a short fragment on the conserved regions, could elucidate high resolution of transmission dynamics during the norovirus outbreaks. Although epidemiological analysis failed to identify specific food items with high risk of symptomatic infections, as norovirus was detected from an index case, a symptomatic food handler, I agree that this is a food-born outbreak caused by norovirus contaminated from food. Please find below minor suggestions to improve the manuscript.

1. I agree that this could be a food-born outbreak, but have you conduct risk analyses with those exposed and unexposed to any food/drink in the hotel? Authors analyzed for individual food/drink items but is there any overall differences in relative risk between those exposed and unexposed to any items? This analysis may enable to confirm if the virus spread by person-person or by food/use of restaurant.

We thank the reviewer for this recommendation but we have not conducted this analysis. Our main hypothesis is that food-handlers could have contributed to the initial spread of the virus

and that subsequently, the outbreak might have been further propagated by both direct person-to-person and direct contact with contaminated surfaces (lines 67-70), as shown by the shape of the epidemiological curve. For clarity, we have included a sentence in the text (line 68-69).

2. There are some food items that showed statistically significant high risk with $p < 0.05$. I would recommend to briefly explain/discuss about these items in the text.

As the reviewer correctly pointed out some food items (fried egg, risotto with mushrooms, mayonnaise, ice drink, surimi and cream cheese) showed a p value < 0.05 . However, since a very low proportion of cases (between 1 to 9 depending on the food item) could be explained, they were not taken into account. We have now included the following sentence as a foot note in supplementary table 1 "These food items had a p value < 0.05 . However, since a very low proportion of cases could be explained they were not taken into account".

3. Please explain what kind of specimens and when they were collected (e.g. days after the onset) during the microbiological investigation.

We have added in the text that stool samples were used for microbiological analyses (line 81 and line 84) as well as the time when they were collected (line 82-3). We have also included the kind of specimen (stool) in Supplementary Table 3.

4. As norovirus typing tool provides both capsid and polymerase type information and discrepancy between capsid and polymerase type already indicates that this strain (GI.5[P4]) is a recombinant virus with GI.5 and GI.P4 virus, I would suggest to omit the simplot and recombination analyses from the manuscript.

We agree and have moved the Simplot analysis to the supplementary data (now supplementary figure 1). We have changed the text in the manuscript accordingly.

5. Please provide legend for the Supplementary Figure 1.

We have added the legend in the Supplementary Figure 1 (now named Supplementary Figure 2).

6. Table S1: Please indicate if the blank (symptom in second part of the outbreak) indicates 0 case or no data available.

We have amended the table indicating that the blank indicates 0.

7. Lines 116-120: I found several typos for GI.5[P4]. e.g., GI5[P4], GI.5 [P4]. No period or extra space before the bracket.

We thank the reviewer for the careful proofreading. These oversights have been corrected.

Reviewer #2:

In this study, researchers investigated instances of norovirus infection among the guests and food-handlers at a hotel in the Summer of 2021. The majority of the samples sequenced was of GI.5P[4]. The authors performed a thorough survey of foods eaten by guests during the outbreak period, but no definitive foodsource was implicated. Transmission by fomite or food-handlers was also a possibility.

1. Out of 16 sequenced samples, 15 were GI.5P[4]. It would be good to mention in the manuscript what genotype the remaining one was.

In fact, only 15 samples were successfully genotyped. The remaining one could not be genotyped. Therefore, we have amended the phrase as follows: "Nucleotide sequencing was successful in 15/16 samples (93.7%). All sequenced strains were genotype GI.5[P4]"

2. Line 64, "The first confirmed case, with symptom onset on July 21, was one food-handler". Looking at Figure 1, this sentence may appear confusing because Figure 1 showed one probable symptomatic client (light brown) and one confirmed subcluster II (light green). The authors referred to "subcluster II" as a genetic grouping, which not introduced until Figure 2. I recommend modifying this sentence to something like "The first known infection occurred on July 21 involving a probable symptomatic client and a symptomatic laboratory-confirmed food-handler." To reduce confusion, the authors may want to add something like "Subsequent genetic analysis of the GI.5P[4] identified the strains as members of subcluster I and II" in the Figure 1 legend.

We thank the reviewer for pointing this out and we have amended the manuscript accordingly to reviewer's suggestions.

3. Figure 1. I counted 127 squares, but the legend mentioned n=110 known onset date and n=8 collection date. Why the discrepancies?

We agree that the legend appears confusing and therefore we have amended it. The legend of figure 1 states that n=110 refers to "probable cases" which are those in light green (n=1) and light brown (n=109). Regarding "confirmed cases" we forgot to indicate in the legend the total

number (n=17). Among those 17 confirmed cases, 8 were asymptomatic food handlers (no illness onset) and therefore we used the sample collection date. To clarify the legend, we have amended as follows: “Probable cases with known illness onset date were included (n=110). For confirmed cases (n=17) we used the illness onset date except for asymptomatic food handlers (n=8) for which sample collection date was used”.

Additionally, there were 11 confirmed food-handlers and 1 client infected with subcluster II, yielding 12 genotyped samples, but only 11 strains shown on the phylogenetic tree?

As stated in line 89, “phylogenetic trees were inferred from study sequences (only those with >450 nucleotides [nt], n=14)”. Among the 15 samples that were positive for GI.5[P4], one sequence (corresponding to subcluster II) was too short for phylogenetic analysis (the PCR is in the junction polymerase/capsid so is better to have long sequences (>450 nt) to infer robust trees with approximately 200 nt for each, polymerase and capsid). Therefore, this sequence was not included in the phylogenetic analysis. It is also indicated in supplementary table 3 that “#Only sequences >450 nt were used for phylogenetic analysis”.

4. Line 79. The authors wrote "A total of 18 samples (from 7 symptomatic and 8 asymptomatic food-handlers, 3 symptomatic clients) were positive for norovirus GI. Supplementary table 3 describes samples for laboratory testing during the outbreak investigation." However, this table indicates that there were only 6 symptomatic food-handlers positive for GI. Why?

We apologise for the oversight, we meant “17 samples (from 6 symptomatic....)”. We have now corrected it.

5. Supplementary Figure 1 is so hazy it's not even legible from the source file. Please ensure the quality of the figure.

We have changed all the files (including supplementary figure 1 now named supplementary figure 2) to TIFF as recommended by the journal: “*Figures: Editable, high-resolution, individual figure files are required at revision, TIFF or EPS files are preferred*”. We have also increased the font size for clarity.

6. Chhabra et al., 2019 established the prototype strains for the purpose of genotyping. It is strongly encouraged that the phylogenetic trees include the necessary prototypic strain (AB042808 Chiba407 for RdRp, and AJ277614 Musgrove for VP1).

Following the reviewer’s recommendation, we have included the reference strains (as Chhabra

et al., 2019 established) indicated with a black triangle in phylogenetic trees (figure 2 and Supplementary figure 2).

7. At least 8 references were missing volumes and page numbers.

We have amended the references with missing data.

January 18, 2023

Dr. Maria Dolores Fernandez-Garcia
Instituto de Salud Carlos III
Centro Nacional de Microbiología
Majadahonda, Madrid
Spain

Re: Spectrum04857-22R1 (Rare Recombinant GI.5[P4] Norovirus causing a large foodborne outbreak of gastroenteritis in a hotel, Spain, 2021)

Dear Dr. Maria Dolores Fernandez-Garcia:

Thank you for submitting your manuscript to Microbiology Spectrum. As you will see your paper is very close to acceptance. Please modify the manuscript along the lines I have recommended. As these revisions are quite minor, I expect that you should be able to turn in the revised paper in less than 30 days, if not sooner.

When submitting the revised version of your paper, please provide (1) point-by-point responses to the issues raised by the reviewers as file type "Response to Reviewers," not in your cover letter, and (2) a PDF file that indicates the changes from the original submission (by highlighting or underlining the changes) as file type "Marked Up Manuscript - For Review Only". Please use this link to submit your revised manuscript. Detailed instructions on submitting your revised paper are below.

Link Not Available

Sincerely,

Gabriel Parra

Editorial comments:

1. Please consider removing "Simplot analysis identified GI.5[P4] as an intergenotype recombinant of GI.4[P4] and GI.5[P5] strains" from the abstract as this information does not provide any additional information. Instead I suggest adding "These recombinant viruses have been identified over the last five years circulating globally, warranting further global surveillance."
2. Please spell "WGS" both in the abstract and in the first use on the main text.
3. The importance section is intended to provide a big picture (a nontechnical explanation) of the significance of the study to the field. Please (i) remove the sentences from lines 42-46, (ii) consider "This study highlights the importance of (i) using whole genome sequencing to assure genetic differentiation of GI noroviruses to track chains of transmission during outbreak investigations" for lines 38-40.
4. Please move "Supplementary table 3 describes samples for laboratory testing during the outbreak investigation" after "onset of symptoms" on line 83.
5. Note that the recombination breakpoint will depend on the sequences used as parental strains. Thus, please revise "breakpoint was identified at the ORF1/ORF2 junction."

Preparing Revision Guidelines

To submit your modified manuscript, log onto the eJP submission site at <https://spectrum.msubmit.net/cgi-bin/main.plex>. Go to Author Tasks and click the appropriate manuscript title to begin the revision process. The information that you entered when you

first submitted the paper will be displayed. Please update the information as necessary. Here are a few examples of required updates that authors must address:

Please return the manuscript within 60 days; if you cannot complete the modification within this time period, please contact me. If you do not wish to modify the manuscript and prefer to submit it to another journal, please notify me of your decision immediately so that the manuscript may be formally withdrawn from consideration by Microbiology Spectrum.

Ref.: Spectrum04857-22

Rare Recombinant GI.5[P4] Norovirus causing a large foodborne outbreak of gastroenteritis in a hotel, Spain, 2021

We have addressed the editorial comments as follows (all the amends on your manuscript are indicated by highlighting in yellow the changes):

Editorial Comments:

1. Please consider removing "Simplot analysis identified GI.5[P4] as an intergenotype recombinant of GI.4[P4] and GI.5[P5] strains" from the abstract as this information does not provide any additional information. Instead I suggest adding "These recombinant viruses have been identified over the last five years circulating globally, warranting further global surveillance."

Done

2. Please spell "WGS" both in the abstract and in the first use on the main text.

Done

3. The importance section is intended to provide a big picture (a nontechnical explanation) of the significance of the study to the field. Please (i) remove the sentences from lines 42-46, (ii) consider "This study highlights the importance of (i) using whole genome sequencing to assure genetic differentiation of GI noroviruses to track chains of transmission during outbreak investigations" for lines 38-40.

Done

4. Please move "Supplementary table 3 describes samples for laboratory testing during the outbreak investigation" after "onset of symptoms" on line 83.

Done

5. Note that the recombination breakpoint will depend on the sequences used as parental

strains. Thus, please revise "breakpoint was identified at the ORF1/ORF2 junction."

We have deleted the phrase

January 20, 2023

Dr. Maria Dolores Fernandez-Garcia
Instituto de Salud Carlos III
Centro Nacional de Microbiología
Majadahonda, Madrid
Spain

Re: Spectrum04857-22R2 (Rare Recombinant GI.5[P4] Norovirus causing a large foodborne outbreak of gastroenteritis in a hotel, Spain, 2021)

Dear Dr. Maria Dolores Fernandez-Garcia:

Your manuscript has been accepted, and I am forwarding it to the ASM Journals Department for publication. You will be notified when your proofs are ready to be viewed.

Sincerely,

Gabriel Parra
Editor, Microbiology Spectrum
